# Antimicrobial Resistance of *Staphylococcus aureus* Isolated between 2017 and 2022 from Infections at a Tertiary Care Hospital in Romania

**DOI:** 10.3390/antibiotics12060974

**Published:** 2023-05-28

**Authors:** Daniela Tălăpan, Andreea-Mihaela Sandu, Alexandru Rafila

**Affiliations:** 1Microbiology Department I, Faculty of Medicine, “Carol Davila” University of Medicine and Pharmacy, 050474 Bucharest, Romania; andreeasandu30@gmail.com (A.-M.S.); alexandru.rafila@umfcd.ro (A.R.); 2“Prof. Dr. Matei Balș” National Institute of Infectious Diseases, 021105 Bucharest, Romania

**Keywords:** resistance, *Staphylococcus aureus*, MRSA, Romania

## Abstract

This study aimed to evaluate the frequency of isolation of *Staphylococcus aureus* from different pathological samples processed in the Microbiology Laboratory of the National Institute of Infectious Diseases “Prof. Dr. Matei Balș”, Romania, between 1 January 2017 and 31 December 2022, aiming to establish the ratio of methicillin-resistant to methicillin-susceptible *Staphylococcus aureus* strains and the antibiotic resistance pattern of isolated microorganisms. The data of isolates originating from routine diagnostic tasks were analyzed retrospectively using laboratory data from the microbiology department. Up to 39.11% of *Staphylococcus aureus* strains were resistant to oxacillin (MRSA), with 49.97% resistance to erythromycin and 36.06% inducible resistance to clindamycin. Resistance rates to ciprofloxacin, rifampicin, gentamicin, and trimethoprim-sulfamethoxazole were 9.98%, 5.38%, 5.95%, and 0.96%, respectively. There was no resistance to vancomycin. Between 2017 and 2022, the percentage of MRSA strains decreased from 41.71% to 33.63%, sharply increasing to 42.42% in 2021 (the year of the COVID-19 pandemic, when the percentage of strains isolated from lower respiratory tract infections was higher than that of strains isolated from wounds or blood, as in previous years). This study showed a high percentage of MRSA strains (39.11% overall) with a higher proportion of these strains isolated from the blood (42.49%) compared to other clinical specimens.

## 1. Introduction

*Staphylococcus aureus*, a Gram-positive bacterium commonly found in the environment, is part of the natural flora of human beings [1]. Most healthy people (over 60%) have this bacterium on their skins and mucous membranes of their upper respiratory tracts, primarily in the nares [2,3]. Approximately 20% of people are long-term carriers [4]. Most of these individuals do not experience any clinical symptoms, as it rarely causes infections if the skin is intact [5]. However, *Staphylococcus aureus* can spread to the bloodstream or internal soft tissues. In that case, it can potentially cause various infectious diseases ranging from minor skin infections and soft tissue infections, such as impetigo, cellulitis, scalded skin syndrome, folliculitis, and abscesses, to severe life-threatening conditions such as fatal pneumonia, osteomyelitis, toxic shock syndrome, endocarditis, and bacteremia [6,7].

It is estimated that infectious diseases are the second most significant cause of mortality globally. The growing danger of drug-resistant microorganisms poses a severe public health problem globally [8]. *Staphylococcus aureus*, which is widely recognized as a significant pathogen in clinical and community environments, is notoriously resistant to penicillin and other antimicrobials [9]. The production of β-lactamase enzymes causes this resistance, with the first report of a penicillin-resistant strain of *Staphylococcus aureus* published in 1945 [10,11]. In 1884, Friedrich Julius Rosenbach first identified this bacterium. However, it was not until the 1930s that enzyme testing was used to detect a staphylococcal infection due to coagulase production by this microorganism. Physicians then began diagnosing and treating *Staphylococcus aureus* using penicillin. Before 1940, 75% of those infected with *Staphylococcus aureus* would die. However, by the end of the 1940s, a resistant strain had developed, causing traditional penicillin to no longer effectively treat the infection [2,12].

The methicillin-resistant *Staphylococcus aureus* (MRSA) is a strain resistant to all penicillin, including methicillin and other narrow-spectrum β-lactamase-resistant penicillin antimicrobials. Moreover, it has been a great challenge to medicine since MRSA causes the same types of infections as other strains of *Staphylococcus aureus* but is resistant to the most common antimicrobials [13]. The rise and dissemination of MRSA, comprising both hospital-associated MRSA (HA-MRSA) and community-associated MRSA (CA-MRSA), is a significant issue on a global scale [14,15]. The emergence of antibiotic resistance in *Staphylococcus aureus* has been attributed mainly to the acquisition of genetic determinants through the horizontal gene transfer of mobile genetic elements [16], the alteration of drug binding sites on molecular targets, and the increased expression of efflux pumps. Conventionally, HA-MRSA has been linked with multidrug resistance and staphylococcal cassette chromosome mec (SCC*mec*) types I, II, and III, while CA-MRSA has been connected to SCC*mec* types IV and V and the presence of Panton-Valentine leukocidin genes. Combinations of inhibitors targeting different sites were used to reduce the probability of resistance arising from mutations [10,14].

Notwithstanding, only some antibiotics with novel chemical classes have been introduced in the past 30 years. Here are some examples of drug classes used to treat *Staphylococcus aureus* infections and their mechanisms of action. Vancomycin, a glycopeptide antibiotic, is extensively used to treat severe infections caused by MRSA strains in hospitalized patients. It binds to the dipeptide D-Ala4-D-Ala5 of lipid II, blocking the transglycosylation and transpeptidation catalyzed by PBP2 (penicillin-binding protein 2) and PBP2a (penicillin-binding protein 2), a protein that is essential to bacterial cell wall synthesis and can prevent peptidoglycan remodeling [9,17]. With vancomycin, up to six gene mutations are required to reduce drug access to the lethal target [10]. Linezolid, an oxazolidinone drug, was approved in the year 2000 for the treatment of challenging HA-MRSA infections. Linezolid is the only wholly synthetic antibiotic that acts on the ribosome. The binding site is in the ribosomal peptidyl transferase center (PTC) in the 50S ribosome subunit, and it impairs the amino-acyl moiety of aa-tRNA, inhibiting peptidyl transferase and peptide bond formation [18,19]. Erythromycin is a macrolide that inhibits the polypeptide exit next to the PTC. Currently, macrolides are not regularly used to treat staphylococcal infections but have a role in *Staphylococcus aureus* infections. Semisynthetic macrolides, such as clarithromycin and azithromycin are used therapeutically to treat bacterial infections caused by microorganisms different from this one. As a result, the commensal staphylococci are regularly exposed to macrolides, which may account for erythromycin resistance being commonly identified in clinical specimens [12,19].

In recent studies, it has been observed that some patients suffering from COVID-19 developed pulmonary bacterial co-infection (identified within 48 h of presentation) and secondary infections (identified after 48 h of admission) or superinfection, which has a negative effect on their prognosis [8,20]. There is considerable variance in the literature regarding the epidemiology of MRSA lung infections in patients with COVID-19, with the relative prevalence ranging from 2% to 29% when all other bacteria are considered and from 11% to 65% when *Staphylococcus aureus* is the common denominator [21,22]. Although various patient-specific environmental factors could be responsible for the predominance of *Staphylococcus aureus* co-infections post-admission in patients with COVID-19, the findings of previous studies suggest that this infection may be partially attributed to the treatment course. Overall, MRSA remains one of the most frequently encountered causative pathogens of pulmonary infections in patients with COVID-19 [20,23,24,25].

Therefore, in this study, we aimed to evaluate the frequency of isolation of *Staphylococcus aureus* from different pathological samples processed in the Microbiology Laboratory of the National Institute of Infectious Diseases “Prof. Dr. Matei Balș”, Romania, between 1 January 2017, and 31 December 2022, to establish the ratio of MRSA strains to methicillin-susceptible *Staphylococcus aureus* strains (MSSA) and the trend in the frequency of isolation of MRSA strains in different clinical specimens and also to monitor the resistance of MRSA/MSSA strains to non-beta-lactam antibiotics.

## 2. Results

### 2.1. The Source of Staphylococcus Aureus Strains

A total of 1672 *Staphylococcus aureus* strains were isolated between 2017 and 2022, with the numbers per clinical specimen being similar between the pre-pandemic years (2017–2019) and reducing during the COVID-19 pandemic and the post-pandemic years (2020–2022). Staphylococci were most commonly isolated from wounds (57.78%), followed by blood (18.72%), lower respiratory tract secretions (9.39%), ocular secretions (8.01%), and then from the urine, ear secretions, pleural fluid, and joint fluid (6.1%) (Table 1).

Over the years, there was no significant difference in the proportion of *Staphylococcus aureus* strains isolated from various clinical specimens, with one exception: in 2021, these bacteria were isolated the most from the lower respiratory tract (LRT, 43.43%; N = 43), compared to wound secretions (25.25%; N = 25), blood (17.17%; N = 17), ocular secretions (4.04%; N = 4), and other clinical specimens (10.10%; N = 10), *p* value < 0.001.

### 2.2. Antimicrobial Susceptibility

A summary of the antimicrobial susceptibility of *Staphylococcus aureus* strains is provided in Figure 1. Up to 39.11% of *Staphylococcus aureus* strains were MRSA, with no strain being resistant to vancomycin but with two of them being resistant to linezolid (0.12%) and three being resistant to teicoplanin (0.18%). The three strains that were resistant to teicoplanin had different minimum inhibitory concentrations (MICs); one had 4 mg/L and two had 8 mg/L, and 18 out of 1672 (1.08%) strains had elevated MIC (2 mg/L). Moreover, 22 out of 1672 (1.32%) strains had vancomycin MIC = 2 mg/L. Among all *Staphylococcus aureus* strains, two were resistant to linezolid (MIC > 4 mg/L), but 17 out of 1672 (1.02%) had a MIC of four, with the rest of them having MICs of ≤2.

The rate of resistance to erythromycin was high (49.97%), and the rate of inducible resistance to clindamycin was 36.06%. The lowest resistance rate (<10%) was that of resistance to ciprofloxacin, moxifloxacin, gentamycin, rifampicin, and trimethoprim-sulfamethoxazole.

The rate of penicillin resistance was high in all strains (>80%), regardless of the clinical specimen (Figure 2). MRSA strains were most commonly isolated from the blood (oxacillin resistance: 42.49%), followed by wound secretions (39.85%) and the lower respiratory tract (37.13%), and, to a lesser extent, from other clinical specimens (ear secretions, urine, pleural fluid, and joint fluid, 29.41%; ocular secretions, 23.88%). *Staphylococcus aureus* strains isolated from ocular secretions were less resistant to all antimicrobials compared to the strains isolated from other clinical specimens, except for gentamicin, where resistance was similar for the strains isolated from wound secretions.

MRSA strains varied over the years (Figure 3), having a steady decrease from 2017 (41.71%) to 2022 (33.63%), with a sharp and significant increase in 2021 compared to 2020 (42.63% vs. 35.35%, *p* < 0.001).

MRSA strains were more resistant to all antimicrobials than MSSA strains (Figure 4) with one exception: linezolid, to which only two MSSA strains were resistant (0.2%). None of the strains were resistant to vancomycin.

MRSA strains showed an increased resistance rate to tetracycline in 2022 compared to 2017 (48.26% vs. 70.27%, *p* = 0.09), fluoroquinolones (ciprofloxacin 10.47% vs. 16.88%, *p* = 0.21; moxifloxacin 6.98% vs. 18.42%, *p* = 0.01), and rifampicin (6.4% vs. 18.42%, *p* = 0.06) (Figure 5); however, the increase was not statistically significant. Decreased resistance to clindamycin (from 83.23% in 2017 to 56.96% in 2022, *p* = 0.88) and erythromycin (from 83.14% to 73.42%, *p* = 0.54) was observed. Resistance to gentamicin and trimethoprim-sulfamethoxazole remained low and variable through the years.

## 3. Discussion

In the CDC’s antibiotic resistance threats report from 2019, MRSA was placed in the “serious threats-public health threats that require prompt and sustained action” category, being responsible for approximately 323,700 infections in hospitalized patients, with an estimated 10,600 deaths in 2017 [26]. In Europe, between 2017 and 2019, the MRSA rate among invasive infections in the European Union according to the European Centre for Disease Prevention and Control antimicrobial resistance surveillance found Romania on top of the list, with 45.4%, 43%, and 46.9%, respectively. Only in 2020 was Romania in second place (after Cyprus, 49.1%) despite having an even higher rate than the previous one (47.3%). In 2021, Romania came third with 41% (after Cyprus, 42.9% and Greece, 41.9%) [27]. Several previous studies have reported the rates of MRSA infections in Romania to range from approximately 30% to 70% [28,29,30,31,32,33].

This study presents evidence of *Staphylococcus aureus* resistance to different antimicrobial agents. Bacterial strains were isolated from samples obtained from infected patients in a tertiary mono-disciplinary hospital, which was declared a COVID-19-dedicated hospital and attended only to patients who tested positive for SARS-CoV-2 virus either by PCR (polymerase chain reaction) or antigen detection in the nasopharyngeal swab from March 2020 to May 2022. From 2017 through 2022, 1672 strains of *Staphylococcus aureus* were isolated. The number of strains isolated during the pre-pandemic years (2017–2019) was higher than those isolated during the COVID-19 pandemic and post-pandemic years (2020–2022). There was no significant distinction in the percentage of isolated strains from different clinical samples, except in 2021 when most isolations were from the lower respiratory tract (43.43%). Our results are similar to those presented by De Santis et al., according to which *Staphylococcus aureus* was most commonly isolated from respiratory samples of patients with COVID-19 (31.1%) [34].

The report summarizes the antimicrobial susceptibility of *Staphylococcus aureus* strains. Approximately 39% of the strains were shown to be MRSA, and no resistance to vancomycin was detected, perhaps because vancomycin treatment in eligible patients in Romania is performed only in the hospital over a short course of two weeks. However, a small percent of strains (22 out of 1672; 1.32%) had vancomycin MIC = 2 mg/L, which is on the border of the wild-type distribution and may be an impaired clinical response if used, according to the European Committee on Antimicrobial Susceptibility Testing (EUCAST) guidelines. It had been feared that MRSA might acquire vancomycin (Van) resistance from enterococci, resulting in untreatable invasive severe infection. Although there have been a few isolated cases of vancomycin-resistant *Staphylococcus aureus* (VRSA), these strains have not spread and have not become a permanent presence in hospitals [35,36].

Two other strains (0.12%) were resistant to linezolid, and three (0.18%) were resistant to teicoplanin. Staphylococcal isolates with reduced susceptibility to glycopeptides, such as vancomycin and teicoplanin, are a significant public health concern because staphylococci are often resistant to various drugs. Glycopeptides are widely used in Europe, where vancomycin is the antibiotic of choice for treating MRSA infections; however, in cases such as endocarditis, osteomyelitis, and septic arthritis, teicoplanin could be considered [37]. Since glycopeptides may be the only remaining effective drugs, initial reports of glycopeptide-resistant staphylococci have caused alarm [38]. The development of resistance to teicoplanin has been documented in cases of MRSA [39]. There have been reports in the literature of MRSA strains that are resistant to teicoplanin but susceptible to vancomycin. Both in vitro and in vivo studies have demonstrated that the MIC of teicoplanin increased 2–16 times; whereas, those for vancomycin only increased by less than two times [37,39,40]. Another study by Majchrzak et al. showed that out of the 600 MRSA strains, 47 (representing 7.83%) were glycopeptide-resistant, and 11 (23.4%) were confirmed to be VRSA. In contrast, the remaining 36 (76.6%) were shown to be resistant only to teicoplanin [41]. Our study also found that out of all *Staphylococcus aureus* strains, only 3 (0.18%) exhibited resistance to teicoplanin, while none exhibited resistance to vancomycin. Teicoplanin resistance has become more prevalent than vancomycin resistance since the initial reports of glycopeptide-resistant staphylococci [42].

Considerable resistance to erythromycin (49.97%) was observed, and the rate of clindamycin-inducible resistance was 36.06%. Ciprofloxacin, moxifloxacin, gentamicin, rifampicin, and trimethoprim-sulfamethoxazole had the lowest resistance rates (all <10%). MRSA strains have experienced a general decline from 2017 (41.71%) to 2022 (33.63%), with a notable spike in 2021 (42.63%), the second year of the COVID-19 pandemic. Moreover, the rate of bacterial secondary infection in patients hospitalized for COVID-19 was high worldwide. In a study conducted in Medellin, Columbia, *Staphylococcus aureus* was the second most isolated microorganism (24%), and in another study conducted in Italy, 40.7% of patients were co-infected with it [36,43]. Per our findings, in 2021, this bacterium was most commonly isolated from the lower respiratory tract (43.43%).

Despite originating from the bacterium, the two strains of *Staphylococcus aureus* (methicillin-resistant and methicillin-susceptible) have distinct resistance and virulence factors, which contribute to determining the type of population affected, their capability to combat traditional treatment methods, and their overall rate of mortality and morbidity [19]. The MRSA strains isolated in this study were more resistant to antimicrobials than the MSSA ones, with the sole exception of linezolid, for which only two MSSA strains (0.2%) were resistant. When introducing linezolid, they asserted it would not be subject to cross-resistance and that resistance would hardly develop. However, evidence of resistance has emerged [20]. There appears to be a correlation between the clinical use of linezolid and a decrease in the MIC of vancomycin in *Staphylococcus aureus,* suggesting that alterations in the clinical application of antibiotics may have an impact on bacterial resistance trends [44].

The rate of penicillin resistance was high in all specimens selected (over 80% of the strains isolated). We found the highest rate of MRSA in blood samples (42.49%), followed by wound secretions (39.85%), the lower respiratory tract (37.13%), and less resistance in the other clinical specimens (such as ear secretions, urine, pleural fluid, and joint fluid—29.41%; ocular secretions—23.88%). *Staphylococcus aureus* strains isolated from ocular secretions had lower resistance to all antimicrobials than those isolated from other specimens, except for gentamycin, for which the resistance was similar to that of the strains isolated from wound secretions. Per the findings of other studies, compared to wound secretions, MSSA-related ocular secretions could be easily cured with regular antibiotics such as erythromycin. Zheng XY et al. observed a correlation between the patient’s age and erythromycin resistance, with topical erythromycin being the most popular over-the-counter antimicrobial drug for common childhood illnesses such as conjunctivitis and bacterial dermatitis [45]. We can attribute an increase in antimicrobial resistance to the misuse of antibiotics (non-prescribed use), incorrect dosage, incorrect duration of treatment, or their use to treat non-bacterial diseases.

In conclusion, it is still possible to effectively treat most *Staphylococcus aureus* infections caused by MRSA by switching drugs or using different combinations. However, it should be noted that the treatment of persistent infections, such as infective endocarditis, is difficult because underlying health conditions weaken the immune system and also because bacteria develop the ability to avoid antibiotics by forming biofilms.

## 4. Materials and Methods

### 4.1. Study Design and Study Setting

This is a retrospective study conducted between 1 January 2017 and 31 December 2022, at the National Institute of Infectious Diseases “Prof. Dr. Matei Balș” in Bucharest, Romania. This facility is a mono-disciplinary tertiary care hospital, which was a COVID-19-dedicated hospital between March 2020 and mid-2022. The study was conducted per the ethical standards of the 1964 Declaration of Helsinki and its later amendments. The institutional review board of the National Institute of Infectious Diseases “Prof. Dr. Matei Balș” granted access to the data without the need for individual informed consent since the data were to be analyzed anonymously. The data were extracted from the hospital’s Microbiology laboratory database.

### 4.2. Bacterial Culture

Between 2017 and 2022, 1672 non-duplicate strains of *Staphylococcus aureus* were isolated from various clinical specimens collected from patients admitted to this institution. Wound secretions, ocular secretions, and ear secretions were collected with sterile cotton swabs. From the lower respiratory tract, sputum, bronchial aspirates, or bronchoalveolar lavage fluid were collected. *Staphylococcus aureus* strains were also isolated from blood, urine, pleural fluid, and joint fluid.

All clinical specimens were sent immediately to the Microbiology laboratory, which works 24/7, for processing. In the laboratory, Gram smears and cultures on appropriate bacterial growth media–Columbia agar with sheep blood (ThermoFisher Scientific™-Oxoid, Wesel, Germany), chocolate agar Polivitex (bioMérieux S.A., Marcy-l’Etoile, France), and lactose agar (CLED, ThermoFisher Scientific™-Oxoid, Wesel, Germany), were performed per the laboratory procedures. Blood culture bottles (bioMérieux FA Plus, FN Plus, SA, SN, and PF plus, bioMérieux S.A., Marcy-l’Etoile, France) were incubated at 37 °C, and those that tested positive per the BacT/Alert (bioMérieux, Inc., Durham, NC, USA) were removed and processed on a 24/7 basis by performing Gram staining and culture on Columbia sheep blood agar, chocolate agar, and lactose agar.

Plates were incubated in aerobic atmospheres at 35 °C ± 1 °C. Growth was observed at 18–24 h.

### 4.3. Staphylococcus aureus Identification 

*Staphylococcus aureus* strains were identified using Matrix-Assisted Laser Desorption Ionization Time-of-Flight mass spectrometry (MALDI-TOF MS), which detects bacterial proteins in whole-cell extracts. Bacterial spectra were analyzed using the Biotyper® software version 3.1 (Bruker Daltonik GmbH, Bremen, Germany).

### 4.4. Antimicrobial Susceptibility Testing

Antimicrobial susceptibility testing (AST) was performed per the EUCAST guideline [46]. MICs were detected using the Sensititre™ system (Thermo Scientific™, Cleveland OH, USA) between 2017 and 2018 and using Romania GP 1, GP 2, and GP 3 EUCAST Micronaut plates (Bruker Daltonics GmbH & Co, KG Bremen, Germany) from 2019 to date (Romania GP 1, 2, and 3 EUCAST cards template are presented in Appendix A and changes in plates from the previous version are marked in bold). Vitek® AST-P592 cards–see Appendix A for the template–(bioMérieux SA, Marcy-l’Etoile, France) were also used through the years if the other system was not available. All plates and cards were used per the manufacturer’s instructions. Antimicrobials tested included oxacillin, penicillin, clindamycin, erythromycin, gentamycin, tetracycline, ciprofloxacin, moxifloxacin, rifampicin, trimethoprim-sulfamethoxazole, linezolid, teicoplanin, and vancomycin.

If there were *Staphylococcus aureus* strains with resistance to teicoplanin or vancomycin, the broth microdilution method using Micronaut’s Vancomycin/Teicoplanin MIC-Strip (MERLIN Diagnostika GmbH, Bornheim-Hersel, Germany) was used to verify the result and find the exact MIC of the strain (Table 2).

The results were interpreted according to EUCAST breakpoint tables for the interpretation of MICs and zone diameters available for each year (version 7.1 to version 12.0) [47]. The breakpoints changed in time for some of the antimicrobials tested: ciprofloxacin (susceptibility breakpoint changed in 2020 to ≤0.001 mg/L, from ≤1 mg/L in 2019), gentamicin (resistance breakpoint changed in 2022 to >2 mg/L, from >1 mg/L in 2021), and rifampicin (resistance breakpoint changed in 2022 to >0.06 mg/L, from >0.5 mg/L in 2021).

Methicillin/oxacillin resistance was detected phenotypically by determining the MIC determination of oxacillin (if >2 mg/L, then are methicillin-resistant) [47] and by the identification of PBP2a via the Penicillin-Binding Protein (PBP2′) latex agglutination test (PBP2′ TEST KIT, Oxoid Limited, Wade Road, Basingstoke, UK) per EUCAST guidelines for the detection of resistance mechanisms [48].

### 4.5. Quality Control

Quality control for AST was performed each time when a lot of Micronaut plates, Vitek® AST P592 cards, or Micronaut’s Vancomycin/Teicoplanin MIC-Strip was changed, with the *Staphylococcus aureus* ATCC 29213 strain, per the manufacturer’s package insert and EUCAST guidelines for internal quality control [49]. The quality control for the PBP2′ TEST KIT was performed per the manufacturer recommendations (for each new lot and weekly thereafter), with a known MSSA (*Staphylococcus aureus* ATCC 292130) and MRSA strain (*Staphylococcus aureus* NCTC 12493–*mec*A positive [49]).

### 4.6. Data Analysis

If *Staphylococcus aureus* strains were isolated from more than one clinical specimen from the same patient, the invasive strains were kept and the ones that caused local infections were eliminated from the analysis to have non-duplicate strains. Data analysis was performed using Microsoft Excel version 16.66.1 (2022 ©Microsoft). The Chi-square test was used to compare population proportions, and results with a *p* value of < 0.01 were considered statistically significant. All *p* values were two-tailed.

## 5. Conclusions

This study demonstrated a high rate of MRSA (39.11% overall), with a higher rate in strains isolated from the blood (42.49%) than in strains isolated from other clinical specimens. With no resistance to vancomycin, we are still confident that infections caused by these bacteria can be treated, even if resistance can emerge, as it happened to teicoplanin and linezolid. While the production of newer drugs is plausible, the implementation of better stewardship practices that may prolong their activity, and more judicious utilization should ensure the continued treatment of many MRSA infections. It is essential to investigate further to maximize clinical treatment results and discover the elements that lead to resistance such as high-risk strains and the molecular genetic makeup responsible for resistance.

## Figures and Tables

**Figure 1 antibiotics-12-00974-f001:**
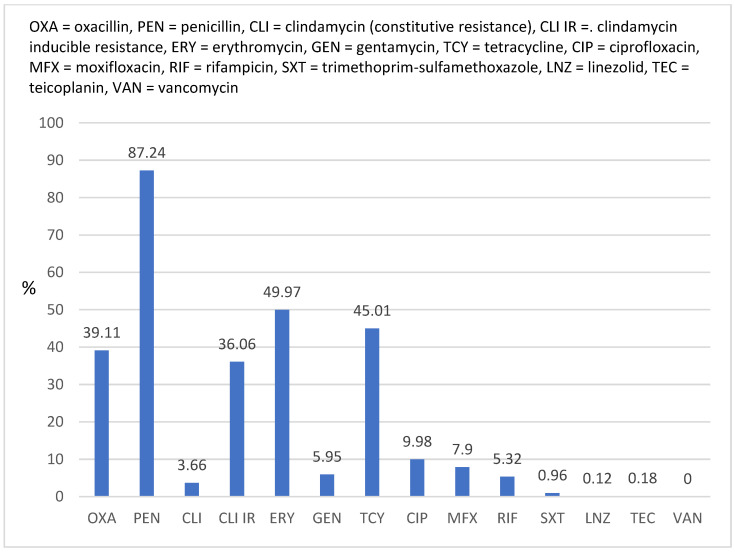
Overall rates of resistance to antimicrobials of *Staphylococcus aureus* strains (N = 1672) isolated between 2017 and 2022.

**Figure 2 antibiotics-12-00974-f002:**
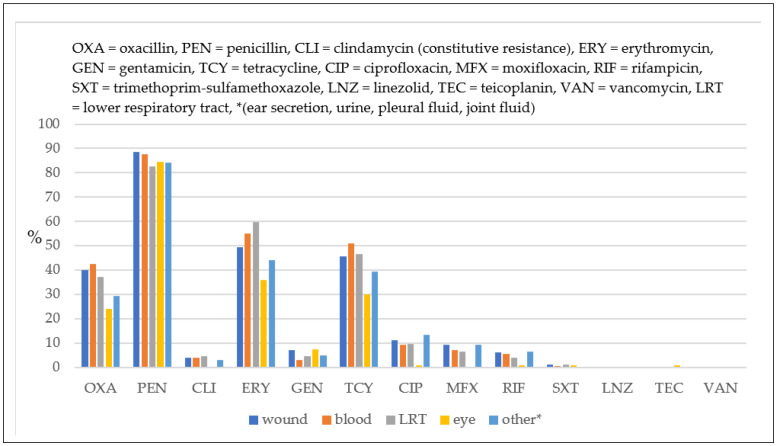
Comparative resistance of *Staphylococcus aureus* strains isolated from different clinical specimens.

**Figure 3 antibiotics-12-00974-f003:**
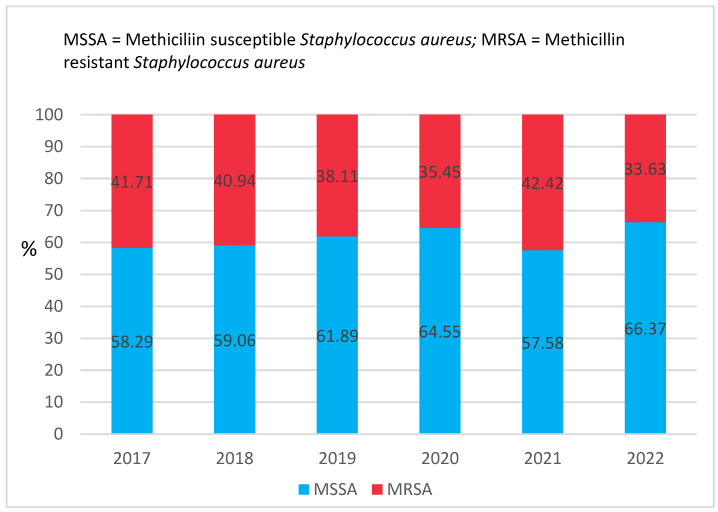
MRSA evolution between 2017 and 2022.

**Figure 4 antibiotics-12-00974-f004:**
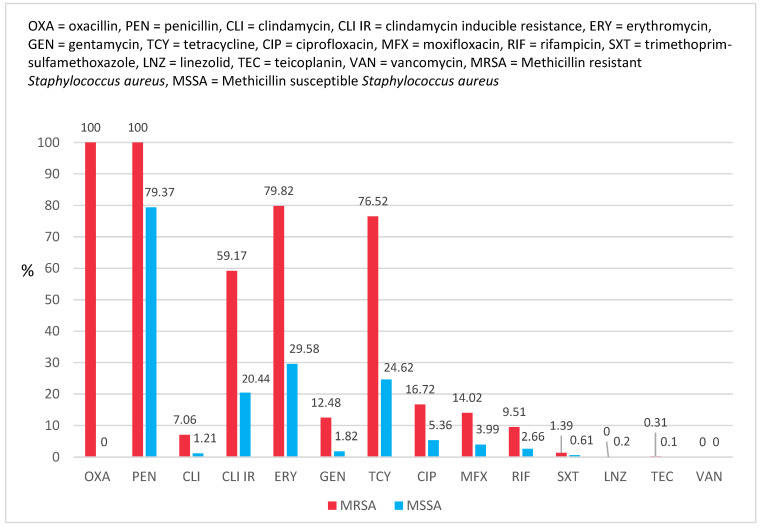
Comparative resistance between methicillin-resistant and methicillin-susceptible *Staphylococcus aureus* strains.

**Figure 5 antibiotics-12-00974-f005:**
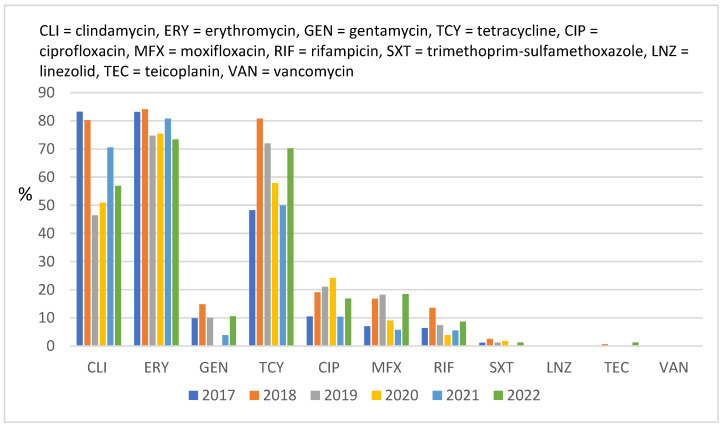
Resistance to non-beta-lactam antimicrobials of MRSA strains isolated between 2017 and 2022.

**Table 1 antibiotics-12-00974-t001:** *Staphylococcus aureus* strains and clinical specimens from which they were isolated.

Year	Wound Secretion (N)	Blood (N)	LRT ^a^ (N)	Eye Secretion (N)	Other ^b^ (N)	Total N (%)
2017	245	68	21	45	31	410 (24.52)
2018	237	75	23	33	13	381 (22.79)
2019	276	87	22	31	30	446 (26.67)
2020	54	27	15	6	8	110 (6.58)
2021	25	17	43	4	10	99 (5.92)
2022	129	39	33	15	10	226 (13.52)
Total N (%)	966 (57.78)	313 (18.72)	157 (9.39)	134 (8.01)	102 (6.1)	1672 (100)

^a^ LRT, lower respiratory tract (sputum, bronchial aspirates, and bronchoalveolar lavage). ^b^ Other (ear secretion, urine, pleural fluid, and joint fluid).

**Table 2 antibiotics-12-00974-t002:** Vancomycin/Teicoplanin MIC-Strip.

1	2	3	4	5	6	7	8	9	10	11	12
GC	VAN 0.25	VAN 0.5	VAN 1	VAN 2	VAN 4	TEC 0.25	TEC 0.5	TEC 1	TEC 2	TEC 4	TEC 8

GC, Growth Control; VAN, vancomycin; TEC, teicoplanin.

## Data Availability

Data are available upon reasonable request to the corresponding author.

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
