# Peer review of "Antimicrobial Resistance of Staphylococcus aureus Isolated between 2017 and 2022 from Infections at a Tertiary Care Hospital in Romania"

_antibiotics, 2023, doi:10.3390/antibiotics12060974_

Round 1
Reviewer 1 Report
Comments to the authors:
1. The manuscript has a few typographical and spacing errors. Therefore, it is essential that the authors thoroughly check for them.
As an example, in section 2.1. The source of Staphylococcus aureus strains, 5th line, “…..the least from urine, eye secretions, pleural and joint fluid (6,1%).” where the word eye secretions should be replaced with ear secretions.
2. Can authors comment on the causes of non-resistance to vancomycin among strains?
3. Why did the authors not investigate the resistance of Staphylococcus aureus isolates in response to various antimicrobial drug combinations?
4. Conclusions can be strengthened.
5. The format of references is inconsistent; authors should maintain uniformity.
Author Response
Dear reviewer,
We are grateful to you for taking time to read our paper. Thank you for all the issues that you found, because resolving them, we made our paper better.
Kind regards,
Daniela Talapan

Reviewer 2 Report
The Introduction is far too expanded.
Lines 15-16: Please rephrase this sentence
Line 19 and elsewhere: gentamycin must read gentamicin (as this antimicrobial agent originates from Micromonospora spp. and not from Streptomyces spp.)
Line 35: Delete “suppose”
Line 66: SCCmec as one word with mec in italics
Figure 3: Susceptibility to methicillin is not correct as the authors have tested the strains for susceptibility to oxacillin.
Figure 4: It cannot be true that you have oxacillin-resistant strains that are penicillin-susceptible.
Figure 5: I am in doubt whether it is possible to have teicoplanin-resistant S. aureus that are vancomycin-susceptible.
Materials and Methods
Line 359 ff: There is no information in Materials and Methods on how MRSA was confirmed. The gold standard is a combination of a genotypic test (mec gene PCR) with a phenotypic test (PBP2a slide agglutination test and/or oxacillin MIC). An oxacillin MIC alone is not sufficient.
Lines 396 ff: The data analysed originate from two different systems: broth microdilution and Vitek automated system. The results from these systems are not directly comparable. The reader also does not get the information, which antimicrobial agents in which testing concentrations are available Rom GP 1, GP 2 and GP 3 EUCAST Micronaut 400 plates as well as the Vitek® AST-P592 cards.
Lines 396 ff: It is compulsory to include suitable quality control strains when performing Antimicrobial Susceptibility Testing (AST). In Material and Methods, there is no information about the included QC strains and the values obtained with them. Without QC information, all data cannot be regarded as valid.
Lines 407-408: Have the breakpoints changed for the single agents during the different years? If so, this should be indicated.
In general, the manuscript might benefit from a thorough revision of the English language by a native speaker.
Author Response
Dear reviewer,
We are grateful to you for taking time to review our paper. Thank you for all the issues that you found, because resolving them, we made our paper better.
Kind regards,
Daniela Talapan

Reviewer 3 Report
Line 19: correct the spelling of gentamicin (and elsewhere e.g. line 405 – check throughout – including figure legends)
Line 31 states that: “Most healthy people have this bacterium on their skin and mucous membranes of the upper respiratory tract”. Colonization of healthy people by Staph. aureus is common but I’d be surprised if “most people” were colonized (i.e. more than 50% of people). Please provide a supporting reference to provide evidence for this statement.
Line 35 “However, suppose Staphylococcus aureus can spread to the bloodstream or internal soft tissues”. This sentence does not make sense. Why should the reader “suppose”?
Methods
Line 377: there is no need to say “blood for blood culture, urine for urine culture” just simply say “blood and urine”……
Line 393: what does “vezi artic cu vet 49” mean?
Results
Line 119: “eye secretions” is listed twice in this sentence.
Line 122 infers that there was a significant increase in LRT infections caused by S. aureus in 2021. This should be analysed using an appropriate statistical method and a ‘P’ vale should be quoted.
Line 200: Again, wherever possible, employ a statistical method i.e. was the change from 2020 to 2021 statistically significant?
More information should be provided regarding the strains showing resistance to teicoplanin and linezolid. Where MICs performed? What were their MIC values for these antibiotics?
I strongly recommend that any resubmitted draft is carefully edited by an expert in English. Improvements are required throughout most of the paper in the use of English.
The text should be reviewed and edited throughout by an expert in English.
Author Response

(The authors gave the same response as above.)

Round 2
Reviewer 1 Report
The manuscript may be considered for publication.
Reviewer 2 Report
Almost all of my initial concerns have been clarified. I am still not convinced that there are really teicoplanin-resistant isolates that are vancomycin-susceptible.
Reviewer 3 Report
The paper, including the use of English, is much improved.